# What is a "Good" figure: Scoring of biomedical data visualization

Hector Torres[1], Efe Ozturk[1,2], Zhou Fang[3,4], Nicholas Zhang[1,3], Shuangyi Cai[1], Neel Sarkar[1], Ahmet F. Coskun[1,2,3,4,5]*

1 Wallace H. Coulter Department of Biomedical Engineering, Georgia Institute of Technology and Emory University, Atlanta, Georgia, United States of America, 2 School of Electrical and Computer Engineering, Georgia Institute of Technology, Atlanta, Georgia, United States of America, 3 Interdisciplinary Bioengineering Graduate Program, Georgia Institute of Technology, Atlanta, Georgia, United States of America, 4 Machine Learning Graduate Program, Georgia Institute of Technology, Atlanta, Georgia, United States of America, 5 Parker H. Petit Institute for Bioengineering and Bioscience, Georgia Institute of Technology, Atlanta, Georgia, United States of America

* ahmet.coskun@bme.gatech.edu

## Abstract

Biomedical data visualization is critical for interpreting complex datasets, yet the clarity and quality of visualizations vary widely across tools and applications. This study introduces a comprehensive framework for evaluating biomedical figures and benchmarking visualization platforms. We developed Metrics for Evaluation and Discretization of Biomedical Visuals using an Iterative Scoring algorithm (M.E.D.V.I.S.), a quantification system that systematically assesses figure quality based on four criteria: Complexity, color usage, whitespace, and the number of distinct visualizations. The algorithm integrates dimensionality reduction, clustering, and thresholding to classify figures and generate tailored feedback for improvement. In parallel, we conducted a comparative analysis of 26 widely used visualization tools, evaluating each based on ease of use, customizability, financial cost, and required background knowledge. To demonstrate real-world applicability, we present case studies on figure evaluation in published research and introduce SpatioView, an interactive, browser-based platform for exploring spatial omics data. Collectively, our findings highlight the need for standardized evaluation methods and provide accessible solutions for improving figure design in biomedical research, education, and industry.

## Introduction

Data visualization serves as a foundational pillar in the communication of scientific ideas, providing essential contextual understanding across a wide range of media including research papers, infographics, presentations, and educational content. Its methodologies and technologies span a broad spectrum, offering both automated visualization creation and customizable options for users seeking greater control. A

**Data availability statement:** Relevant data and analysis results are available at https://figshare.com/projects/Biomedical_Data_Visualization_and_Scoring/196741 The codes are available at https://github.com/coskunlab/M.E.D.V.I.S-Algorithm.

**Funding:** AFC holds a Career Award at the Scientific Interface from Burroughs Wellcome Fund and a Bernie-Marcus Early-Career Professorship. A. F. C. was supported by start-up funds from the Georgia Institute of Technology and Emory University. Research reported in this study was supported by the National Institute of General Medical Sciences of the National Institutes of Health under award number T32GM142616.

**Competing interests:** The authors have declared that no competing interests exist.

vast array of software tools and knowledge sources exist to guide users in crafting sophisticated data visualizations. However, these tools differ substantially in customizability, usability, and suitability for domain-specific applications such as biomedical engineering. Accordingly, it is crucial to examine not only the functionalities of these tools but also the guiding principles that underpin their use and their applicability to biomedical datasets.

In establishing the foundational aspects of data visualization, this study critically examines key literature, including two texts dedicated specifically to biomedical data visualization [1,2] and a third that focuses more broadly on visualization aesthetics and design [3]. From these sources, several central principles emerge: visualization type selection, appropriate tool choice, integration of statistical interpretation, and alignment with specific medical applications [4]. Visualization type selection involves choosing the most appropriate graphical format to convey the underlying message of the data. Tool selection pertains to identifying suitable platforms for data handling, analysis, and presentation. The reviewed texts advocate the use of platforms like Microsoft Excel, Python, Tableau, and R Studio, which provide a range of software-based and code-based options. Furthermore, they explore distinctions among these platforms in terms of flexibility, user expertise required, and optimal use cases [1,2].

Equally emphasized is the necessity of statistical literacy in visualization design. Understanding descriptive and inferential statistics, along with significance testing, is positioned as essential for accurate data communication. The medical applications discussed in these texts span a wide range of domains, including genomics, healthcare analytics, and machine learning for disease prediction and diagnosis [1–3,5–7]. Design principles such as color theory, Gestalt principles, and shape-based encoding are also given substantial attention, particularly in biomedical contexts. These texts caution against the overuse of three-dimensional visualizations, noting that such elements can detract from message clarity and hinder interpretation [1,2,8,9]. Importantly, the texts also differ in their depth of coverage—while some provide detailed recommendations for visualization tool selection, others offer only generalized guidance on desirable software features [3].

Beyond textbooks, an increasing number of online resources contribute meaningfully to the discourse on biomedical visualization. These include tutorials and opinion pieces that focus on figure layout, clarity, negative space, accessibility (e.g., color blindness), and aesthetics [10–22]. Such resources serve as valuable supplements, particularly for beginners or those without access to comprehensive textbooks.

Biomedical engineers operate across diverse sectors—academic, industrial, and clinical—each of which imposes unique demands on data visualization. In industry, visualization is often employed to demonstrate product efficacy, validate performance metrics, and communicate outcomes to non-specialist stakeholders, including investors and consumers. In contrast, academic biomedical visualization emphasizes clarity of communication for scientific discovery, peer review, and reproducibility. Academic use cases are less focused on marketing and more oriented toward user comprehension and functional transparency in experimental design and outcomes (Fig 1).

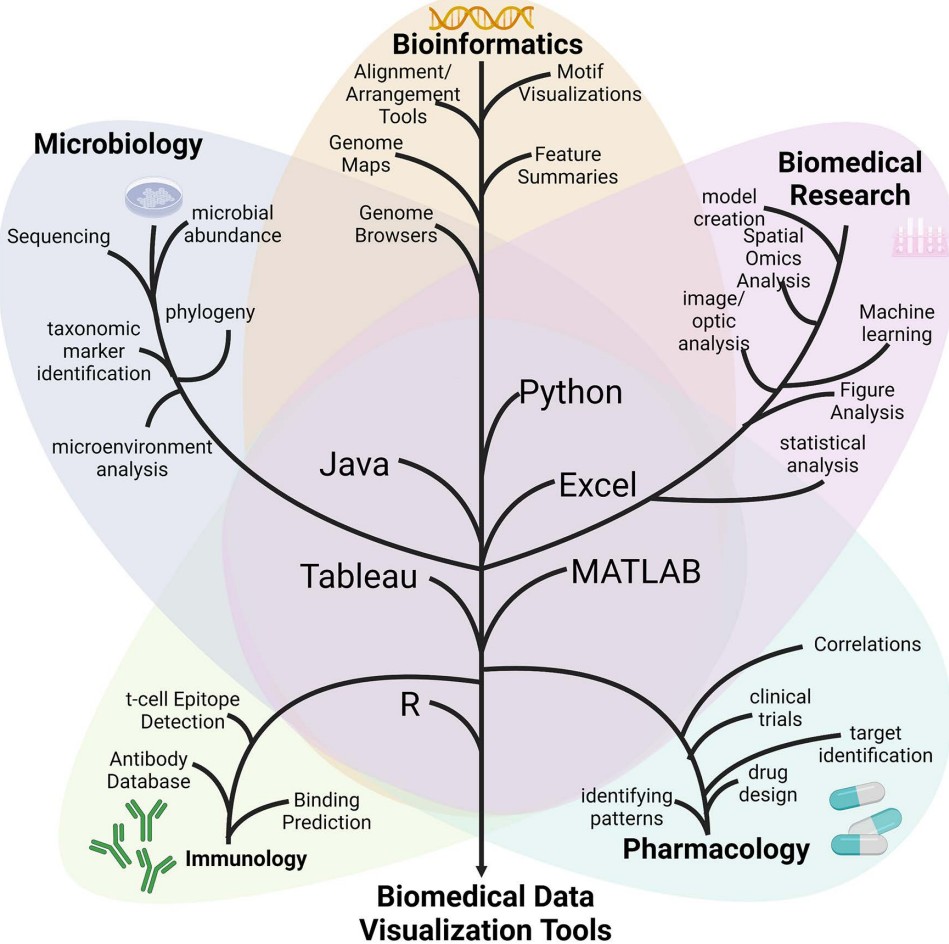

**Fig 1. Visual representation of different areas of biomedical engineering.** The subareas in each area, and the associated tools they commonly use. In each Major area, all the sub-areas are listed as common items that are studied in the major areas. All of the sub-areas are connected to the major area by the color and are also funneled down and connected to the tools area in the middle. All the tools connect down to the final output of Biomedical Data Visualization Tools. Created with BioRender.com.

Despite the significance of biomedical data visualization, there remains a notable gap in how it is taught, particularly within formal educational curricula. Most data visualization courses and textbooks prioritize general-purpose data science methods, often neglecting the specialized needs of biomedical applications. Effective biomedical visualization requires not only a firm grounding in statistical and computational tools but also the capacity to translate complex biological data into clear, interpretable visual narratives [23]. Unfortunately, many educational resources do not emphasize this interdisciplinary challenge, leading to gaps in both training and practice.

Textbooks and syllabi often reflect this disconnect. Among seventeen textbooks analyzed, only six focus explicitly on biomedical data visualization, while the remaining eleven approach visualization from broader perspectives such as business, IT, or sociology. Many of these texts are authored by individuals without biomedical training, though their content occasionally addresses healthcare statistics, bioinformatics, or biomedical image processing [1,2]. This diversity of authorship underscores the need for more targeted educational resources for biomedical engineering students and practitioners. Similarly, syllabi often recommend the use of general tools such as Python, R, Tableau, and Microsoft Excel, which provide a strong foundation but lack specific emphasis on biomedical datasets and their unique requirements [24].

Visualization strategies in biomedical applications also vary by data type and domain. For example, scatter plots and line graphs are ideal for displaying continuous variables such as symptom progression over time, allowing for interpolation across patient samples (S1 Fig). Conversely, geographic heat maps are more effective for spatially distributed statistics such as mortality rates by region. Thus, choosing the correct visualization type is highly dependent on data context and communication goals [1].

In research settings, data visualization is used to interrogate and summarize findings, whether through simple bar plots or complex models such as neural networks and spatial visualizations. For instance, image-based data displays can model proteomic data in 3D environments to facilitate the exploration of molecular structures [25]. In industry, visualization not only supports scientific claims from clinical trials but also functions as a persuasive tool for stakeholders. Public health agencies like the Centers for Disease Control and Prevention (CDC) routinely use data visualizations to communicate trends in health outcomes, such as infant mortality or vaccine coverage, thereby driving policy and public awareness.

Despite the centrality of data visualization in biomedical research, there remains a critical gap in standardized frameworks for evaluating figure quality. While general best practices exist, they are often qualitative, inconsistent, or poorly suited to the specific demands of biomedical datasets. This study addresses this gap by introducing an objective, algorithmic method for assessing biomedical figures based on measurable criteria including complexity, color use, whitespace, and the number of visualizations. The proposed methodology, termed M.E.D.V.I.S., applies machine learning and dimensionality reduction to score and classify figures. The framework is validated through multiple case studies, including quantitative analysis of published figures, comparative visualization using different software tools, and the creation of an interactive online platform named SpatioView for visual exploration of high-dimensional spatial omics data. Our findings demonstrate that M.E.D.V.I.S. can reliably differentiate high- from low-quality figures, provide actionable feedback, and support the creation of more effective biomedical visualizations. This work lays the foundation for future standardization in figure evaluation and visualization design within the biomedical sciences.

## Results

### Evaluation of biomedical data visualization tools

To assess the usability and suitability of data visualization software for biomedical applications, we conducted a comparative evaluation of 26 tools based on four key criteria: Ease of use, customizability, required background knowledge, and financial cost. These criteria were chosen to reflect the diverse needs of biomedical users, from researchers creating figures for publication to educators and developers building interactive data visualizations.

Each software tool was rated on a 5-point scale (The high score means better performance) for each criterion, with definitions adjusted to suit the nature of the metric. For customizability, the rating was based on the number of distinct visualization types a tool could produce. Variants of the same chart (e.g., 2D vs. 3D bar charts) were not counted as separate types. Ease of use was operationalized as the time (in seconds) required to create a basic histogram from an uploaded dataset, measured using standardized tasks. Financial cost reflected the monthly subscription price, with exceptions made for tools that required quotations or offered both free and paid versions. Required background knowledge was qualitatively assessed based on whether the software offered a "quick start" guide or required coding proficiency.

Notably, tools such as *Visual.ly* and *Infogram*—primarily designed for infographics and simple charts—scored highly in ease of use but lacked flexibility in visualization types. In contrast, tools like *Tableau* and *IBM Watson* demonstrated stronger performance across both ease of use and customizability (**Table 1**) [26,27]. Coding-based platforms such as *Jupyter/Python*, *Plotly*, and *MATLAB* offered extensive customization and data handling capabilities but scored lower in ease of use due to their steeper learning curves and reliance on programming knowledge.

**Table 1. A comparison of 26 different visualization software tools using specific criteria.**

| Name | Financial Cost (per month) | Required Knowledge | Ease of use | Customizability |
|---|---|---|---|---|
| Tableau | 2 and Free | None | 5 | 4 |
| Dundas BI | 1 | None | 4 | 1 |
| Jupyter/Python | Free | Coding principles, Matplotlib, Seaborn, etc. | 1 | 2 |
| Zoho Reports | 1 and 3 | None | 5 | 2 |
| Google Charts | Free | Javascript syntax | 5 | 2 |
| Visual.ly | Requires a quote | Unknown | 5 | 1 |
| RAW | Free | None | 3 | 2 |
| IBM Watson | Free and 3 | None | 4 | 3 |
| Sisense | Requires a quote | None | 4 | 3 |
| Plotly | Free | Python coding Language | 1 | 3 |
| Data Wrapper | Free and 4 | None | 5 | 1 |
| Highcharts | 1 | None | 5 | 4 |
| FusionCharts | 1 and 3 | Javascript | 5 | 4 |
| Power BI | 1 | None | 5 | 2 |
| QlikView | 1 | None | 4 | 2 |
| Infogram | Free and 1 | None | 4 | 1 |
| ChartBlocks | Free | None | 4 | 3 |
| D3.js | Free | Javascript | 1 | 2 |
| Chart.js | Free | HTML | 1 | 2 |
| Grafana | Free | None | 3 | 2 |
| Chartist.js | Free | None | 2 | 1 |
| Sigma.js | Free | Javascript | 1 | 1 |
| Polymaps | Free | Javascript | 2 | 1 |
| Microsoft Excel | 1 | None | 4 | 3 |
| Google Sheets | Free | None | 4 | 2 |
| MATLAB | 2 | Basic coding Principles | 2 | 5 |

All of the scores of all of the visualization tools were examined across all four criteria of ease of use, financial cost, customizability, and required background knowledge. The high score means better performance in 5-point scale.

## Performance across software categories

All tools evaluated were capable of importing external datasets (*e.g.*, from Excel files), which supports interoperability with biomedical workflows. Tools varied significantly in their processing speed and visualization breadth. For instance, spreadsheet-based tools like *Microsoft Excel* and *Google Sheets* performed well in terms of usability and accessibility but were limited in visual diversity and integration with statistical pipelines. Conversely, platforms like *D3.js* and *Python/Seaborn* allowed for intricate visual design and statistical overlays, though at the cost of user-friendliness.

The comparative results across the criteria show that while many tools cluster around moderate performance levels, certain platforms trade off ease of use for depth of control, and vice versa. Interestingly, coding-based tools, despite higher setup times, consistently scored the highest in customizability (median score: 4.5), whereas GUI-based tools clustered around 2–3. The inverse trend was observed for ease of use (Fig 2).

To evaluate tool performance under realistic biomedical conditions, we tested each software on a large multi-variable dataset representing synthetic clinical data. Standard plots (*e.g.*, histograms, bar charts, scatter plots) were generated using both the software's default charting interfaces and, where applicable, custom scripts. In coding environments, visualization scripts were written manually to replicate common figure types. The quality, responsiveness, and feature

 

### a Customizability Score

| Rating | Definition |
|---|---|
| 1 | 0-5 visuals |
| 2 | 6-10 visuals |
| 3 | 11-15 visuals |
| 4 | 16-20 visuals |
| 5 | 20+ visuals |

### b Ease of Use Score

| Rating | Definition |
|---|---|
| 1 | 120 seconds + |
| 2 | 91 – 120 seconds |
| 3 | 61 – 90 seconds |
| 4 | 31-60 seconds |
| 5 | 0-30 seconds |

### c Scoring Comparison

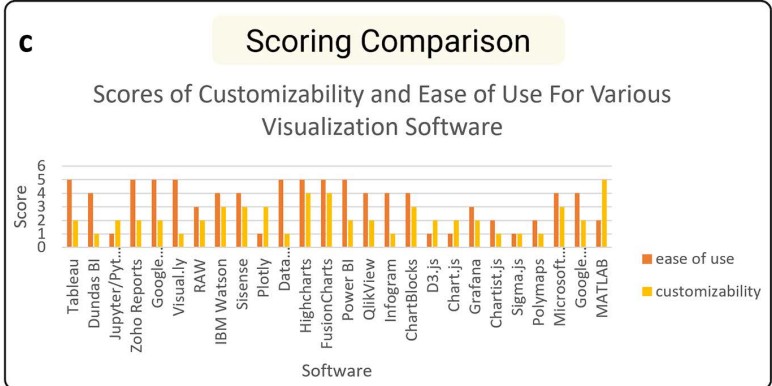

**Fig 2. A Comparison of visualization tool scores and their metrics. (a)** The Rating definition for the Customizability Score is based on the ability of the software to create distinct visualizations. **(b)** The rating definition of the ease of use criterion is based on the amount of time that it takes a user to create a basic histogram from data upload to visualization creation. **(c)** The rating definition of the financial score is defined as the amount that software costs a user financially per month. **(d)** A visual bar graph of the more quantifiable scores for all twenty-six software for comparison among their customizability and their ease of use.

flexibility of each resulting figure were recorded and reviewed. All sample visualizations are included in the Supporting Information section.

The evaluation highlights that no single tool universally outperforms others across all criteria. Rather, each tool offers particular strengths depending on user needs. Graphical user interface (GUI)-based platforms are ideal for quick, accessible visualizations with minimal learning time, making them suitable for early-stage education, exploratory analysis, or public-facing content [26,27]. Coding-based environments, while requiring more expertise, offer significant advantages for precision, automation, and integration with biomedical data analysis workflows.

We recommend that biomedical researchers choose visualization tools that align with their intended audience, technical background, and data complexity. Budget considerations also play a role, as higher-tier software licenses correlate

with expanded functionality. Ultimately, this comparative evaluation offers a practical reference for selecting tools that best support high-quality, reproducible biomedical data visualizations.

## Case study 1: Figure quality analysis using the M.E.D.V.I.S. algorithm

To evaluate the quality of scientific figures in a standardized and reproducible manner, we developed and tested a scoring algorithm, M.E.D.V.I.S., across a set of real-world biomedical figures. In this context, we define a "visualization" as an individual image, graph, or chart within a figure and a "figure" as a composite entity that may include one or multiple visualizations.

Each figure in the dataset was quantitatively evaluated across four predefined criteria: (1) Amount of white space, (2) number of visualizations, (3) color density, and (4) image complexity. These criteria were selected based on design principles commonly emphasized in biomedical data visualization [1–4,28–31].

To examine the robustness of M.E.D.V.I.S. scoring, we designed a weight sensitivity analysis framework in which the relative contributions of complexity, color usage, whitespace ratio, and number of visualizations were systematically varied in ±10% increments while holding the other three metrics constant. Across all perturbations (n = 81), rankings remained highly stable (median correlation with baseline 0.9995, range 0.9981–1.0000). No classification flips were observed (median 0; max 0), and the median mean absolute rank shift was 0.19. These results indicate that the scoring system is resilient to moderate variations in metric weights, which shows both the near-zero divergence from the baseline ranking and the small rank shifts across all weight sets (S2 Fig).

Figures were scored on each dimension and subsequently embedded into lower-dimensional spaces using three dimensionality reduction techniques: Principal Component Analysis (PCA), Uniform Manifold Approximation and Projection (UMAP), and t-distributed Stochastic Neighbor Embedding (t-SNE). This transformation allowed for a more intuitive clustering of figures in a lower-dimensional space and supported comparative analysis of figure attributes.

Following dimensionality reduction, K-Means clustering (with k = 4) was applied independently to each projection to identify inherent groupings in the data. To account for the variability introduced by different embedding methods, a consensus clustering approach was employed. Each figure was assigned a cluster label ranging from 0 to 3, reflecting degrees of agreement across the three clustering outputs. These consensus scores were then binarized: clusters 0 and 1 were labeled as "good" and clusters 2 and 3 as "not good" (Fig 3).

In addition to cluster-based classification, figures were further analyzed relative to quantile-based thresholds to provide criterion-specific feedback. For each metric, the threshold was defined using either the first or third quartile of the distribution, depending on whether higher or lower values were considered favorable. For example, a figure with more than three visualizations (above the first quartile) was flagged for excessive visual content (Fig 4). This allowed the system to generate interpretive advice tailored to each figure, such as reducing whitespace, improving color differentiation, or simplifying layout complexity.

Gauge charts were generated for each figure and each metric, visualizing the score relative to the threshold and providing actionable insights for figure improvement. All scoring outputs and suggestions were saved in accompanying text files for user access (Fig 4).

The algorithm was implemented using Jupyter Notebook in Python. It requires as input: (1) image files of the figures and (2) corresponding captions. Captions must include visual identifiers (e.g., labeled subpanels like "(A)" or "(b)"), which the algorithm uses to estimate the number of visualizations per figure. Improperly formatted captions were the primary source of error during testing and must be standardized for optimal performance.

To test the tool, a diverse set of figures submitted by external authors was analyzed. While most figure components were correctly parsed, inaccuracies in determining the number of visualizations occasionally arose due to irregular caption formatting. As a workaround, the algorithm relies on explicit subpanel references to estimate the visualization count.

To validate the algorithm's figure-counting accuracy, we conducted a chi-squared test comparing the algorithm's output to human-annotated labels across a subset of figures. Results indicated strong agreement for the majority of inputs, confirming the method's reliability for structured captions.

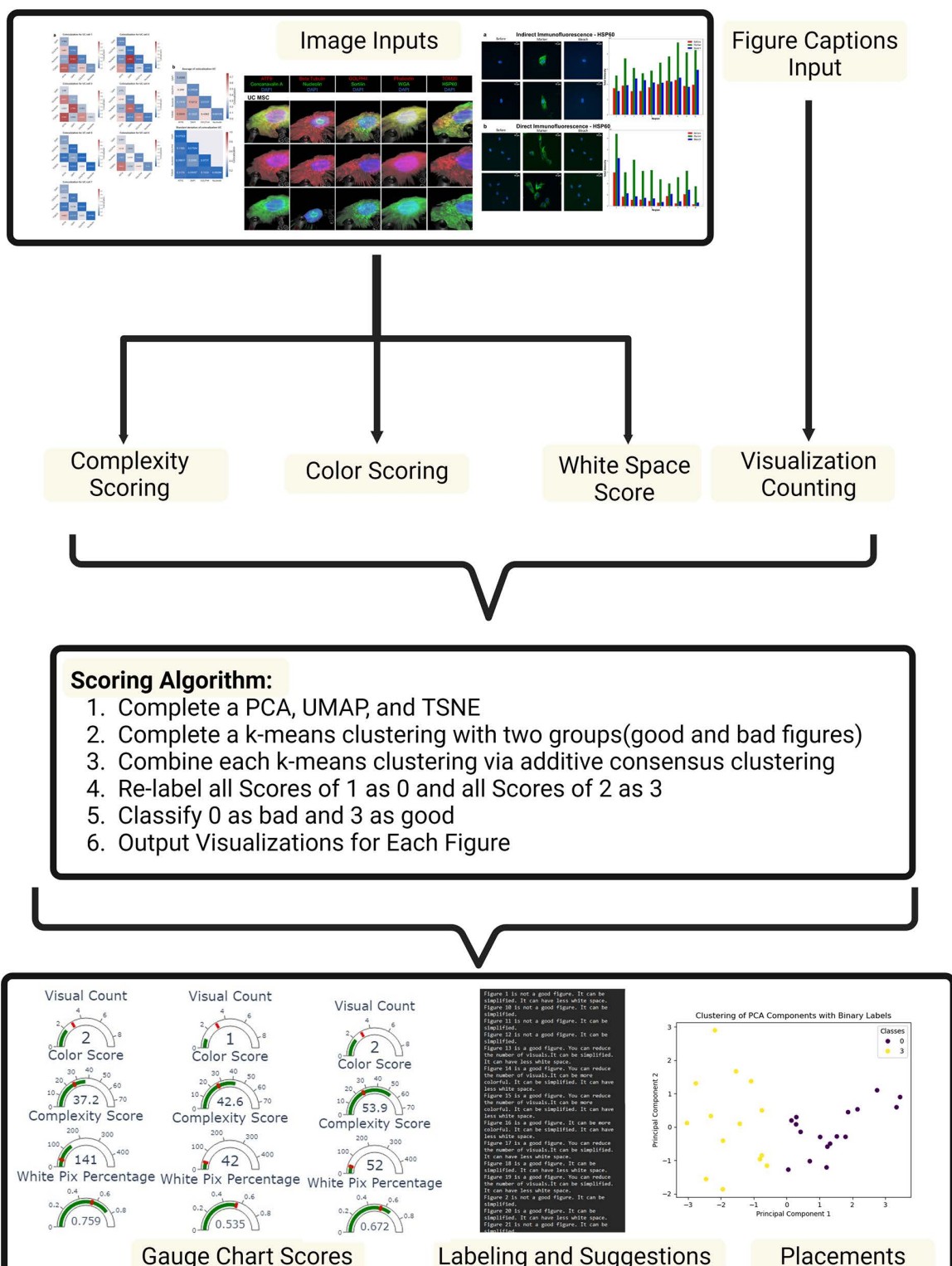

**Fig 3. The process of using the M.E.D.V.I.S. algorithm.** The images and their captions are scored by the algorithm based on complexity (segmentation), color scoring (difference in RGB channels), whitespace (white pixel count), and visualization count (caption indication). All are placed into three dimensionality reduction methods, which are then processed in a k-means clustering algorithm with k = 2 to represent good and bad figures. The

individual clustering events are then combined using an additive consensus clustering method to classify figures as either "good" or "not good" compared to the other terms. The algorithm also packaged all visualization and data into different folders for the user to peruse. Created with BioRender.com.

However, several limitations emerged. First, the clustering and scoring are relative, meaning that figures must be assessed in groups. A single figure analyzed in isolation will lack sufficient context to be categorized as "good" or "bad," and no comparative clustering can be performed. Second, figures without captions or with unlabeled visualizations are automatically excluded from accurate evaluation of visualization count, potentially skewing final scores.

Interestingly, even among figures labeled as "good," some failed individual metrics such as white space or complexity. One figure, for example, passed on visual count, color density, and complexity but exceeded the threshold for unused white space. In contrast, another figure was labeled "not good" despite favorable scores on color and count, suggesting that the criteria may not contribute equally to the final label. This finding motivates further refinement through weight optimization of each metric in future versions of M.E.D.V.I.S. (Fig 4).

## Case study 2: Comparative visualization of a spatial genomic dataset

The visualization of data is often influenced by user preferences, yet the choice of visualization method can significantly impact the narrative conveyed by the data [32–36]. This study sought to explore the varying representations of the same dataset across different software platforms (S3 Fig). To prevent software crashes, a subset of a genomic dataset was utilized, containing x, y, and z positional coordinates, cell identifiers, and mRNA gene information. This dataset was provided and processed under the guidance of a research team member.

The process involved selecting data for visualization, organizing it by cell identifiers to isolate genes for each cell, and calculating the percentage of each gene for visualization purposes (Fig 5). this process was specifically for pie charts, similar procedures were carried out for bar charts, histograms, and violin charts, among others.

Several visualization libraries in Python, Java, and R were employed, alongside Tableau, to assess their capabilities on the same dataset. The visualizations produced by each platform varied aesthetically, with differing levels of processing required. Notably, Tableau required less processing effort compared to custom code in the coding languages, suggesting the potential advantages of visualization software for quicker visualization creation (S4 Fig).

Furthermore, the dataset was used to visualize "good" and "bad" figures to illustrate common issues and best practices in data visualization. "Bad" figures exhibited challenging viewpoints, misplaced axes, unlabeled axes, inadequate use of color, and excessive white space, while "good" figures adhered to best practices including appropriate color usage, clear data presentation, proper axis labeling, and effective spacing [3] (S5–S7 Figs).

This study highlights the importance of software selection and visualization design in accurately conveying the meaning of data. Limitations included restricted access to certain software and the large size of the dataset, which necessitated data reduction without compromising information integrity. Overall, the study underscores that software choice and visualization type significantly influence the utility and interpretation of visualizations.

## Case study 3: SpatioView – An interactive visualization platform for spatial omics data

The objective of this case study is to develop an interactive viewer for high-dimensional datasets, enabling researchers to quickly visualize their data without the need for custom scripts. This tool, named SpatioView, aims to expedite the data exploration process, allowing researchers to efficiently review and validate their results or proceed to further analysis if satisfied. SpatioView was first developed using the Python language and then moved over to HTML using a custom-made website. To enhance user interactivity, PyScript, a method for embedding Python code into an HTML environment, was employed. This approach enables dynamic visualizations [37] that change appearance upon hovering over data points, offering users a more engaging experience. Additionally, users have the option to render static versions of the interactive graphs for direct use in publications.

# Example Gauge Chart Outputs

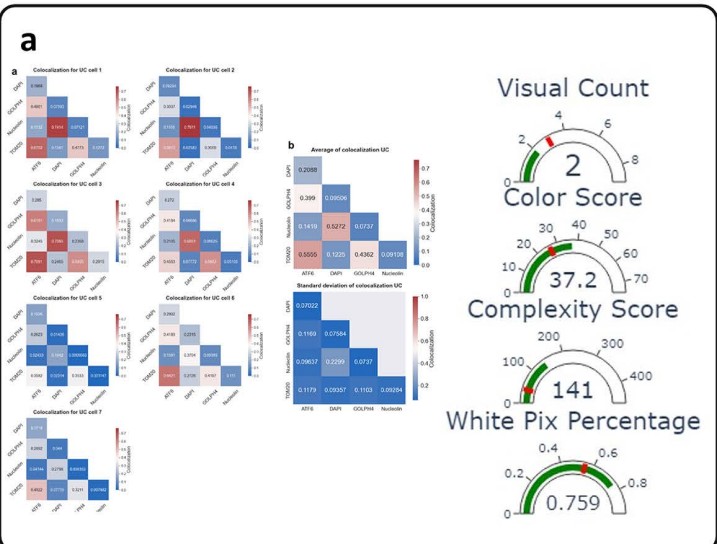

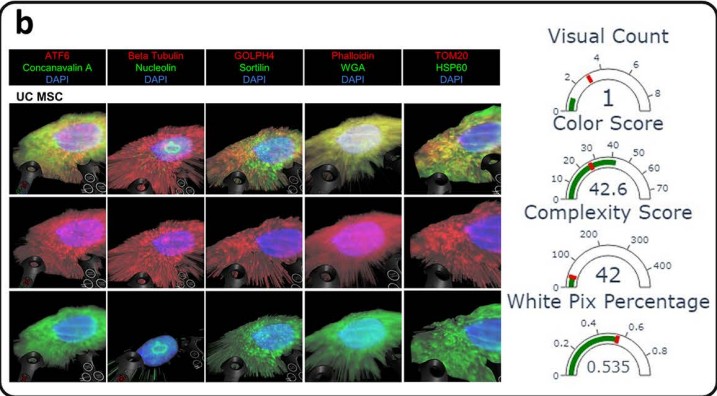

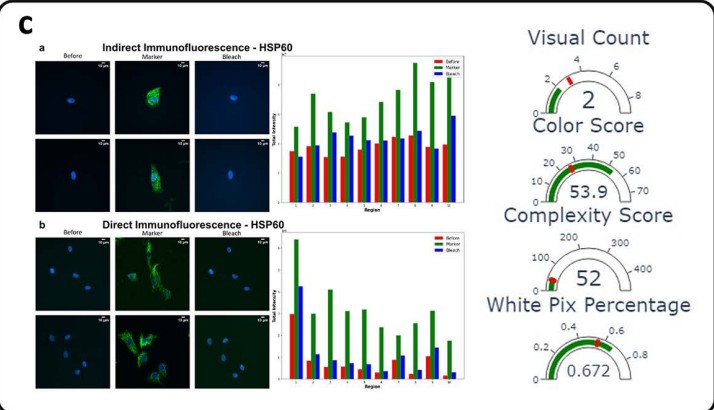

**Fig 4. Example gauge chart outputs produced by the M.E.D.V.I.S. algorithm.** These gauge charts display each figure's score in four key categories, with the red mark indicating the threshold for acceptable performance. The corresponding images for each gauge chart are shown alongside. **(a)** This figure is considered moderately effective by the algorithm. While it scores well in visual count and color usage, it has high complexity and inefficient white space usage, placing it near the midpoint overall. **(b)** This figure is rated highly by the algorithm. It features low complexity, appropriate white space usage, a limited number of visual elements, and a strong color score. **(c)** This figure performs well in all categories except white space usage, where it slightly exceeds the threshold. Despite this, it is still considered a good figure under the scoring algorithm. Created with BioRender.com.

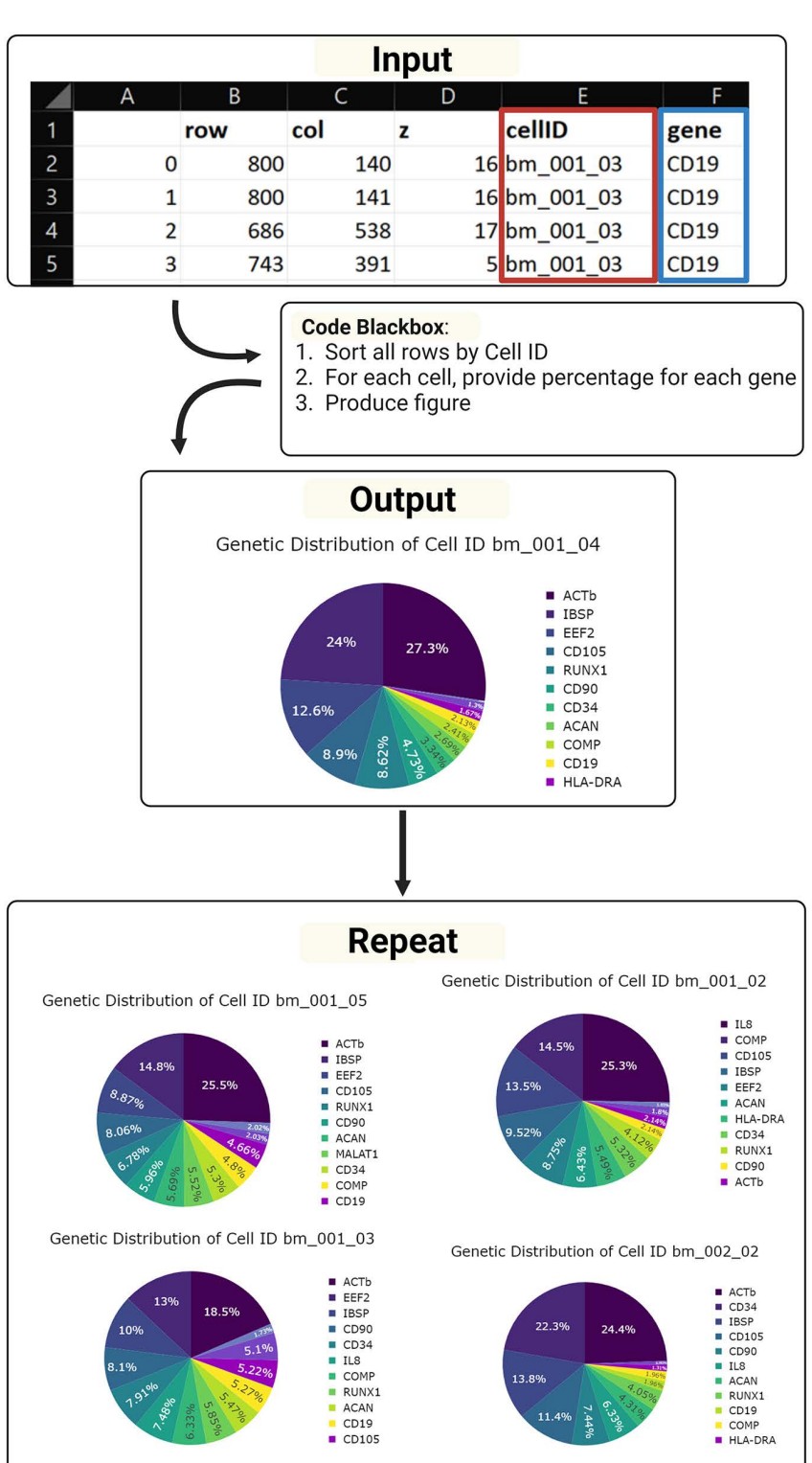

**Fig 5. Cellular data inputs and outputs.** Visualizations from the code were created by taking in a spreadsheet, sorting the cells, and taking the percentage of the number of times any given mRNA appeared in that cell compared to the number of total mRNAs found. This was done for all cells used. Created with BioRender.com.

To facilitate a user-friendly interface, a single-cell spatial multi-omics dataset tracking the behavior of over four thousand fibroblasts under various stimulation levels was utilized. This dataset includes columns detailing the relative levels of p65 RNA transcript and protein in the cytosol and nucleus, where a value of zero indicates low levels and one indicates maximal levels. Notably, the activation of p65 occurs specifically in the nucleus, contrasting the conventional view of protein activation in the cytosol.

Before implementing the website, several Python libraries were employed to compare the visualization quality of different dimensionality reduction algorithms, including PCA, t-SNE, UMAP, and Diffmap. These algorithms yielded visuals with varying levels of interpretability. While some, like PCA, produced easily discernible shapes that corresponded well with data columns, others, such as UMAP, generated shapes that were less intuitive and would not be suitable for publication without additional context. To address this variability, SpatioView allows users to select their preferred visualization method (Fig 6).

We demonstrated SpatioView with high-resolution screenshots (Fig 6c) from a representative spatial transcriptomics dataset containing over 3000 cells [38]. The interface now displays key user-selectable features, including dimensionality reduction method, hover-based cell information, and interactive filtering options. Labels have been added to the figure to indicate control panels, legend placement, and zoom functions. These updates provide a clearer visual narrative of how SpatioView facilitates exploratory data analysis for complex multi-omics datasets.

Despite its promise, SpatioView has limitations that warrant further research and development. The runtime of the interactive visualizer was prolonged, primarily due to the complex rendering of dimensionality reduction methods. Furthermore, UMAP and Diffmap could not be directly rendered on SpatioView due to PyScript's restriction on importing libraries that are not purely Python-based. Nevertheless, the website demonstrates the potential for handling complex spatial multi-omics data effectively [39].

To assess initial alignment with human judgment, we conducted a small mock evaluation study with nine biomedical researchers and graduate students who used M.E.D.V.I.S. on a test dataset. Participants rated each figure as either *Good* or *Bad*, and their consensus labels were compared with the automated M.E.D.V.I.S. classifications. Human raters and the algorithm agreed on 85% of figures, with balanced precision and recall across both categories (S8 Fig). Qualitative feedback emphasized the usefulness of automated feedback for identifying problematic figures and highlighted the potential of M.E.D.V.I.S. to provide reliable, objective guidance on figure quality. While these findings are preliminary and illustrative, they suggest that M.E.D.V.I.S. can effectively approximate expert judgment and may serve as a valuable tool for improving figure design in research and teaching contexts. Future iterations of this case study could involve providing users with access to various graph types and a GUI to customize their visualizations post-processing. Additionally, enabling users to include custom scripts for separate code development and greater data customizability would be beneficial. The website's code could also be enhanced to allow users to select which data columns to display, increasing its generalizability. While these goals were not achieved due to various constraints, unlimited resources could facilitate their implementation.

## Discussion

### Challenges in biomedical data visualization

Despite the growing centrality of data visualization in biomedical research, education, and industry, several persistent challenges must be addressed to ensure the creation of accurate, interpretable, and reproducible figures. A primary challenge lies in the requirement for coding proficiency among figure creators. Tools such as Python and MATLAB offer powerful customization and integration capabilities, but they impose a steep learning curve, particularly for users without formal programming training. Conversely, while spreadsheet-based and GUI tools offer greater accessibility, they can still present usability barriers and lack the fine-grained control and automation afforded by coding environments.

Another critical issue is the subjectivity inherent in figure design. The choices made during visualization—such as axis scaling, color usage, and data selection—are influenced by the creator's interpretation and intent [40]. This subjectivity

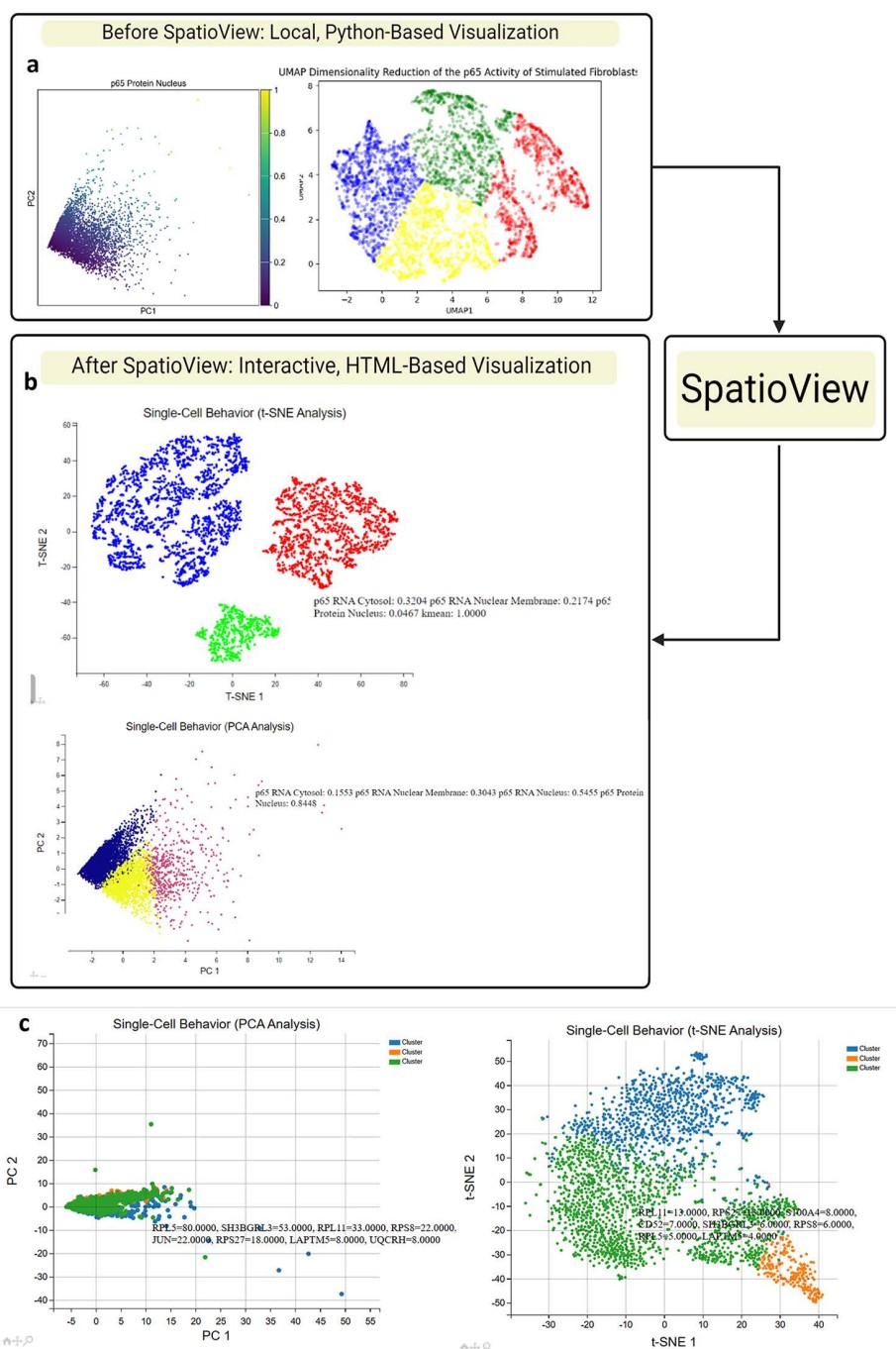

**Fig 6. A comparison between the static Python-based visualizations of the p65 activity in activated fibroblasts and the dynamic, HTML-based visualizations that use these same reduction methods. (a)** UMAP, t-SNE, PCA, and Diffmap were first generated using the Python libraries Scikit-learn, UMAP, and PyDiffmap within Jupyter to generate static graphs as a starting point. Then, Plotly was used as a tentative interactive tool to generate information about each point in an external window. **(b)** After SpatioView: due to PyScript constraints, PCA and t-SNE were implemented as viable dimensionality reduction methods in an HTML-based interface. Interactive features include hover tooltips displaying cell metadata, legend control, and zoom functionality, enabling rapid exploration of cell- and cluster-level properties. **(c)** High-resolution screenshots from a representative spatial transcriptomics dataset (~3,000 cells) illustrate the SpatioView interface, with labeled control panels, legends, and zoom options. These features demonstrate how SpatioView facilitates intuitive, user-driven exploration of complex multi-omics datasets.

introduces the potential for unintentional bias or misrepresentation, particularly in contexts like industry or public communication, where figures may be used to persuade rather than inform. Additionally, readers may misinterpret poorly designed or cluttered figures, further complicating the communication of scientific findings.

Reproducibility poses another major challenge. In coding environments, reproducibility is more straightforward because the figure generation pipeline can be preserved and shared. However, in GUI-based software, where visualizations are often generated through manual steps, recreating a figure with identical design choices is far less reliable. This undermines transparency and makes peer validation more difficult.

Addressing these challenges requires deliberate attention to training, tool design, and evaluation standards. By improving access to visualization education, enhancing the usability of scientific software, reducing subjective design choices through algorithmic guidance, and encouraging reproducibility through code sharing, the biomedical community can significantly elevate the quality and impact of its data visualizations.

### Implications of the M.E.D.V.I.S. framework

The M.E.D.V.I.S. algorithm was developed as a first step toward standardizing the evaluation of figure quality in biomedical contexts. By scoring figures across four quantifiable criteria, comprising complexity, color use, whitespace, and number of visualizations, M.E.D.V.I.S. introduces objectivity into what is traditionally a subjective process. Dimensionality reduction and clustering techniques allowed for the classification of figures into "good" and "not good" categories, while quartile-based thresholding generated actionable feedback for individual metrics. The inclusion of gauge charts and text-based suggestions enables authors to iteratively improve their figures based on standardized metrics.

The weight sensitivity analysis framework provides a structured approach for refining the influence of individual scoring metrics and for validating model outputs against expert consensus. Although we have not yet implemented full-scale optimization, the methodology is designed to be scalable for larger figure repositories.

Importantly, M.E.D.V.I.S. does not replace human judgment but augments it. Our results demonstrate that even figures categorized as "good" may fail specific design criteria, highlighting the value of multidimensional scoring. Conversely, figures classified as "not good" may still contain strong design elements, suggesting the need for future work to optimize the weighting of scoring criteria. Nevertheless, the tool offers a replicable, scalable foundation for assessing figure quality, particularly when applied to full sets of manuscript figures.

### Reflections on case studies and tool comparisons

The case studies presented in this work illustrate the practical and interpretive value of figure design across contexts. Case Study 1 showed that M.E.D.V.I.S. could distinguish between figures with strong visual communication and those lacking clarity or efficiency, offering tailored feedback based on real figure samples. Case Study 2 demonstrated how different visualization platforms produce markedly different outputs from the same dataset, underscoring the need for careful tool selection. In particular, GUI-based tools such as Tableau offered speed and usability, whereas code-based libraries demanded more expertise but offered superior flexibility and control.

Case Study 3 introduced SpatioView, an interactive browser-based viewer for high-dimensional spatial omics data. This platform exemplifies a solution to one of the major challenges in biomedical visualization: accessibility. By eliminating the need for local scripting, SpatioView lowers the barrier for exploratory analysis while offering publication-quality visuals. Limitations such as PyScript's inability to render certain libraries (*e.g.*, UMAP, Diffmap) point to ongoing technical hurdles, but the prototype demonstrates that interactivity and usability can coexist in modern biomedical visualization tools.

Preliminary mock feedback from a small group of researchers indicated strong alignment between human evaluations and the automated classifications produced by M.E.D.V.I.S., as well as positive impressions of SpatioView's potential for rapid data exploration. These findings, while limited in scope, reinforce the potential of the tools to support figure quality assessment and to enhance visualization workflows in research and teaching contexts.

## Limitations and future directions

This study has several limitations. First, the M.E.D.V.I.S. framework relies on relative comparison across multiple figures and cannot evaluate single figures in isolation. Second, while the current scoring criteria were selected based on expert knowledge and common visualization principles, their relative contributions to the final "goodness" score have not yet been optimized through large-scale validation. Third, although our dataset included a broad range of figure types and visualization tools, further validation on larger and more diverse figure repositories will be necessary to generalize the findings.

For SpatioView, limitations included restricted support for libraries with C programming-based dependencies and slower rendering for large datasets. Future versions could incorporate precompiled JavaScript libraries or back-end rendering services to overcome these limitations. Similarly, expanding M.E.D.V.I.S. to incorporate accessibility metrics, such as color blindness compatibility or font size readability, could offer a more comprehensive assessment of figure quality.

## Toward standardization in biomedical visualization

This work underscores the urgent need for standardized evaluation frameworks in biomedical data visualization. As scientific figures grow more complex and datasets become increasingly high-dimensional, researchers must adopt new tools and best practices to ensure that visual communication keeps pace. M.E.D.V.I.S. provides a foundation for automated figure scoring and feedback, while tools like SpatioView illustrate how interactivity can be democratized through accessible platforms. Together, these contributions help move the field toward reproducibility, clarity, and consistency in scientific communication.

# Conclusion

This study underscores the central and evolving role of data visualization in biomedical science. Biomedical data visualization encompasses a wide spectrum of chart types, design principles, and software tools, each tailored to specific applications in research, education, and industry. As the complexity of biomedical datasets continues to grow, so too does the demand for visualization methods that are not only informative but also accessible, reproducible, and interpretable.

In response to these demands, we developed and tested two complementary tools: the M.E.D.V.I.S. algorithm, which provides an objective scoring framework for evaluating the quality of scientific figures, and SpatioView, an interactive web-based platform for exploring high-dimensional spatial omics data. Together, these tools support users in both assessing and improving their visualizations, while promoting best practices in design and communication.

Our findings demonstrate that visualization tools vary significantly in their usability and performance, and that figure quality can be quantified using well-defined visual criteria. By providing automated scoring and interpretive feedback, M.E.D.V.I.S. enables users to iteratively refine their figures. Meanwhile, SpatioView showcases how modern web technologies can lower technical barriers and improve accessibility for non-coding users [41,42].

Ultimately, this work contributes to the foundation for more standardized, transparent, and effective data visualization practices in biomedicine. As visualization becomes increasingly integral to biomedical discovery and dissemination, tools that support evaluation, interactivity, and reproducibility will be critical for advancing the field.

# Supporting information

**S1 Fig. A collection of visual features and their extremes.** This figure illustrates different visual elements assessed by the M.E.D.V.I.S. algorithm, along with examples representing the two ends of each spectrum. (a) Color usage: the top visualization uses full color, while the bottom is presented in black and white. (b) Data type: Examples range from structured spreadsheet data to unstructured image data, illustrating the diversity of input types. (c) Readability: the top figure has low readability, especially in the axes, whereas the bottom figure demonstrates improved clarity. (d) Graph type: Two distinct visualization styles— a summative pie chart and a descriptive heat map—represent the range of graph types possible with the same data. (e) Dimensionality: This panel shows how data can be represented in different dimensions, with

both 2D and 3D plots depicted. (f) Complexity: The top figure shows a simple graph with minimal visual elements, while the bottom figure displays a more complex visualization containing multiple elements. Created with BioRender.com.
(TIF)

**S2 Fig. Weight sensitivity of M.E.D.V.I.S. scoring.** (a) Histogram of ranking divergence values $(1 - corr) \times 10^3$ across all weight perturbations ($n = 81$). Vertical dashed and dotted lines indicate the median and 95th percentile, respectively. (b) Scatter plot of ranking divergence versus mean absolute rank shift, with bubble size encoding the number of classification flips relative to baseline. Dashed lines mark medians, and shaded regions indicate 95th-percentile stability zones.
(TIF)

**S3 Fig. Various depictions of figures can be created using different software using the same data.** (a) A Pie Chart and a bar chart that was created using Tableau. (b) Network and Bubble Charts were created using a Java-based environment. (c) Depictions of a bar chart and a heat map that were created using R Studio. (d) Visualizations created using Plotly and Matplotlib.pyplot libraries in Python. Created with BioRender.com.
(TIF)

**S4 Fig. A depiction of the distribution of information gathered for visualization along with representations of the distribution of Z positional information across all mRNA locations across all cells.** (a) Depicts the five dimensions of the data used including the X, Y, and Z positional data; the name of the mRNA at that position, and the cell ID number where it came from. (b) The Distribution of the Z positional data across all bone marrow cells produced in R. (c) The Distribution of the Z positional data across all bone marrow cells produced in Python. (d) The Distribution of the Z positional data across all bone marrow cells produced in Java. Created with BioRender.com.
(TIF)

**S5 Fig. A series of good and not-good figures.** (a) A depiction of different heat maps depicting the relative frequencies of X and Y positions of various genes across the cellular microenvironment. The leftmost figure depicts easily distinguishable variations in positional frequency and depicts the scale clearly while the rightmost heatmap is difficult to distinguish different regions due to the color and does not have a scale for intensity. (b) A depiction of the overall genetic distribution of cells across X and Y positions. The topmost figure has smaller markers, a title, and appropriate text sizes while the bottommost figure has larger markers making it difficult to distinguish regions of gene markers. (c) 3D Charts that depict the x, y, and z position. The leftmost graph is good as it also depicts a fine gradient for the intensity of the z position whereas the rightmost visual is much more colorful and is difficult to easily see that same information. Created with BioRender.com.
(TIF)

**S6 Fig. Visualizations depicting the outputs of the scores from the scoring algorithm visually for the maximum and minimum of the counts, the complexity, and the color score.** If multiple maximum or minimum were found, only the first maximum or minimum is depicted. (a) The minimum and maximum determined visualization count. (b) The minimum and maximum complexity score. (c) The minimum and maximum color score. (d) Binary representations of what is considered a "bad" and a "good" figure. Created with BioRender.com.
(TIF)

**S7 Fig. A depiction of both the minimum and maximum number of white pixels along with the locations where the scoring algorithm determined that there were white pixels as outlined in red.** If multiple maxima or minima were found, then only the first of each was used in this visualization by the scoring algorithm. (a) The minimum and maximum determined amounts of white pixels found. (b) A Red marking of all of the white pixels that were found by the scoring algorithm. Created with BioRender.com.
(TIF)

**S8 Fig. Comparison of human consensus ratings and M.E.D.V.I.S. algorithm classifications for figure quality.**
(a) Confusion matrix showing agreement between human majority votes (*Good* vs. *Bad*) and algorithm predictions. (b) Per-figure lollipop plot of the proportion of human raters voting *Good* (horizontal axis), with points colored green when the algorithm agreed and red when it disagreed. The shaded region indicates the decision threshold (≥0.5 = *Good*), and right-side annotations show the algorithm's assigned label for each figure.
(TIF)

**S1 File. M.E.D.V.I.S. instruction manual.** This manual provides detailed setup and usage instructions, including file organization, required Python libraries, output folder descriptions, and optional validation steps for automated visualization counts.
(DOCX)

## Author contributions

**Conceptualization:** Hector Torres, Nicholas Zhang, Ahmet F. Coskun.

**Data curation:** Hector Torres, Zhou Fang, Shuangyi Cai, Neel Sarkar.

**Formal analysis:** Hector Torres.

**Investigation:** Hector Torres, Neel Sarkar.

**Methodology:** Hector Torres, Efe Ozturk.

**Project administration:** Ahmet F. Coskun.

**Resources:** Nicholas Zhang.

**Software:** Hector Torres, Efe Ozturk, Neel Sarkar.

**Supervision:** Ahmet F. Coskun.

**Validation:** Hector Torres, Neel Sarkar.

**Visualization:** Hector Torres, Efe Ozturk.

**Writing – original draft:** Hector Torres, Neel Sarkar.

**Writing – review & editing:** Hector Torres, Efe Ozturk.

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
