## [Decision Letter · Decision Letter 0]

24 Apr 2025

Dear Dr. Coskun,

Thank you for submitting your manuscript to PLOS ONE. After careful consideration, we feel that it has merit but does not fully meet PLOS ONE’s publication criteria as it currently stands. Therefore, we invite you to submit a revised version of the manuscript that addresses the points raised during the review process.

We look forward to receiving your revised manuscript.

Kind regards,

Ashad Kabir, PhD

Academic Editor

PLOS ONE

Journal Requirements:

4. Please ensure that you refer to Figure 1 in your text as, if accepted, production will need this reference to link the reader to the figure.

5. Please remove your figures from within your manuscript file, leaving only the individual TIFF/EPS image files, uploaded separately. These will be automatically included in the reviewers’ PDF**.**

**6.** Please include captions for your Supporting Information files at the end of your manuscript, and update any in-text citations to match accordingly. Please see our Supporting Information guidelines for more information: http://journals.plos.org/plosone/s/supporting-information.

Additional Editor Comments :

The reviewers have raised several major concerns. Please carefully consider all comments and ensure that each one is thoroughly addressed in revision.

Reviewers' comments:

Reviewer's Responses to Questions

**Comments to the Author**

1. Is the manuscript technically sound, and do the data support the conclusions?

Reviewer #1: Yes

Reviewer #2: No

Reviewer #3: No

2. Has the statistical analysis been performed appropriately and rigorously?

Reviewer #1: Yes

Reviewer #2: No

Reviewer #3: No

3. Have the authors made all data underlying the findings in their manuscript fully available?

Reviewer #1: Yes

Reviewer #2: No

Reviewer #3: No

4. Is the manuscript presented in an intelligible fashion and written in standard English?

Reviewer #1: Yes

Reviewer #2: No

Reviewer #3: No

Reviewer #1: The topic is very interesting and necessary because currently there are many software for data visualization but limited articles with comparisons on them.

The introduction is too short, the authors should provide other research articles on data visualization (although limited but there are if you google), then draw the novelty of this study.

Figures quality are low, so I cannot read some of the texts in Figure 3 and 4. Please improve the quality.

Line numbers should be added.

Reviewer #2: 1. The abstract should present to the reader in an objective way: Objective, Methodology, Findings, and the relevance of the research. I suggest that you reconstruct the abstract focussing on the points presented;

2. The introduction does not contextualise the subject, it does not present the problem, the objectives, a summary of the methodology, the main findings.

3. The article is not structured as a scientific article, but as a summarised version of a book chapter.

I suggest that the authors reorganise the article and resubmit it at a later date.

Reviewer #3: This study seeks to explore the diverse applications of biomedical visualization in various societal contexts and propose an evaluation framework for scoring visualizations using a bespoke algorithm. Furthermore, this research aims to present an example of interactive data visualizations for research-oriented datasets, achieved through both the scoring of research-based figures and the development of a fully interactive web-based visualization platform.

The motivation behind this study appears to be compelling; however, the overall structure of the paper lacks clarity, making it difficult to follow. It would be beneficial if the authors aligned the figures more closely with the corresponding text to enhance readability and coherence.

Furthermore, the structure of the paper deviates notably from conventional norms. For instance, the introduction is too brief to adequately convey the core ideas underpinning the work; no related literature is reviewed; and it is difficult to discern the principal components of the proposed strategy or the rationale behind the introduced metrics.

**Do you want your identity to be public for this peer review?** For information about this choice, including consent withdrawal, please see our Privacy Policy

Reviewer #1: No

Reviewer #2: No

Reviewer #3: No

---

## [Author Response · Author response to Decision Letter 1]

19 Jun 2025

Rebuttal Letter

Manuscript Title: What is a “Good” Figure: Scoring of Biomedical Data Visualization

Manuscript Number: PONE-D-25-05417

Dear Reviewers,

We thank the editor and reviewers for their thoughtful evaluation of our manuscript. We are grateful for the opportunity to revise and resubmit our work, and we have made extensive improvements to address each point raised. Below, we provide detailed responses to all reviewer comments. We believe that the revised manuscript now meets the expectations for clarity, structure, rigor, and presentation outlined by the journal.

Editorial and Journal Requirements

1. PLOS ONE formatting and structure requirements:

We have revised the manuscript to fully comply with PLOS ONE formatting guidelines, including the correct title page structure, file naming, and placement of figure files as separate TIFFs. Figure 1 is now explicitly cited in the main text, and Supporting Information captions have been added to the end of the manuscript.

2. Code and data availability:

We confirm that all code related to the M.E.D.V.I.S. algorithm and SpatioView platform has been made publicly available via GitHub, and the datasets used for case studies have been shared through Figshare, as noted in the Data Availability Statement.

3. Author affiliations:

Each author has been explicitly linked to their institutional affiliation(s), and any secondary or current affiliations are marked as per PLOS ONE guidance.

Reviewer #1

Comment 1: The topic is very interesting and necessary because currently there are many software for data visualization but limited articles with comparisons on them.

Response: We appreciate the positive assessment and fully agree with the reviewer’s view on the relevance of comparative evaluations in visualization software.

Comment 2: The introduction is too short; authors should provide other research articles on data visualization and clarify the novelty.

Response: We have significantly expanded the Introduction to include relevant literature on biomedical visualization principles [1–4], challenges [5–17], and applications in education and industry. The revised version clearly states the problem, novelty, methodology, and key findings.

Comment 3: Figures 3 and 4 are low quality and hard to read.

Response: Figures 3 and 4 have been re-rendered at 600 DPI resolution and saved as high-quality TIFF files. Labels, font sizes, and contrast have been improved for readability.

Comment 4: Line numbers should be added.

Response: Line numbers have been added to the manuscript file as requested.

Reviewer #2

Comment 1: The abstract should be structured around Objective, Methodology, Findings, and Relevance.

Response: We have fully rewritten the abstract to follow this structure. The revised abstract now clearly outlines the study’s motivation, the development of M.E.D.V.I.S., the tool comparison, case study findings, and broader implications for the field.

Comment 2: The introduction lacks contextualization, a problem statement, a methodology summary, and key findings.

Response: The Introduction has been substantially revised and now includes a clear background on biomedical visualization, the motivation for our scoring framework, a summary of the M.E.D.V.I.S. algorithm, tool evaluation methods, and major results from the case studies.

Comment 3: The article reads more like a book chapter and is not structured as a scientific article.

Response: We have completely reorganized the manuscript to follow the standard scientific structure: Introduction, Results (with subsections for each case study), Discussion (including challenges, implications, limitations, and future work), and Conclusion. Each section now contains clearly defined objectives and key takeaways.

Reviewer #3

Comment 1: The study has compelling motivation, but the paper lacks clarity in structure.

Response: The manuscript has been restructured and rewritten throughout to improve clarity. Each major component—M.E.D.V.I.S., tool comparison, and SpatioView—is now presented as an independent but connected result within the broader context of biomedical visualization needs.

Comment 2: Figures should be aligned more closely with text for better readability.

Response: We have ensured that each figure is now clearly referenced in the text near the paragraph that describes it.

Comment 3: The introduction is too brief and does not review related literature or explain the rationale for metrics.

Response: The revised Introduction includes relevant references on figure design, visualization principles, and evaluation challenges. The rationale for choosing our four scoring metrics (complexity, color, whitespace, visual count) and how they relate to existing design heuristics are now explained in the Results section.

Additional Improvements Noted in Revision

• A new Discussion section was added to synthesize challenges, interpret findings from all three case studies, and describe future directions.

• The Conclusion was rewritten to summarize contributions and emphasize the role of standardization in biomedical visualization.

• The Supporting Information section now includes clear captions and updated figure references.

• All figure files were removed from the manuscript text and uploaded as individual high-resolution files per journal guidelines.

We sincerely thank the reviewers for their insightful comments. Their suggestions have led to a stronger, clearer, and more rigorous manuscript. We hope that the revised version will meet the standards of PLOS ONE and be considered favorably for publication.

Sincerely,

Ahmet F. Coskun, Ph.D.

(on behalf of all co-authors)

ahmet.coskun@bme.gatech.edu

Georgia Institute of Technology

---

## [Decision Letter · Decision Letter 1]

11 Aug 2025

Dear Dr. Coskun,

Thank you for submitting your manuscript to PLOS ONE. After careful consideration, we feel that it has merit but does not fully meet PLOS ONE’s publication criteria as it currently stands. Therefore, we invite you to submit a revised version of the manuscript that addresses the points raised during the review process.<h3 style="margin-right: 0cm; margin-left: 0cm; font-size: 12pt; font-family: 宋体; color: rgb(0, 0, 0); text-align: justify;"><samp>This paper proposes a new algorithmic framework, M.E.D.V.I.S., which quantifies the quality of biomedical images using four metrics: complexity, color usage, white space, and number of visualizations. It then compares and evaluates this framework against 26 commonly used visualization tools. The authors also developed an interactive visualization platform, SpatioView, to enhance the presentation of spatial omics data, emphasizing the importance of standardization and reproducibility in image design.</samp><o:p></o:p></h3><h3 style="margin-right: 0cm; margin-left: 0cm; font-size: 12pt; font-family: 宋体; color: rgb(0, 0, 0); text-align: justify;"><samp>Minor revision suggestions:</samp><o:p></o:p></h3><h3 style="margin-right: 0cm; margin-left: 0cm; font-size: 12pt; font-family: 宋体; color: rgb(0, 0, 0); text-align: justify;"><samp>1. Further quantify the rationality of scoring weights.Although four scoring metrics have been proposed, the weights of each metric for the final “good/bad” classification are not yet clear. It is recommended to add weight sensitivity analysis or expert scoring comparisons to enhance the persuasiveness of the evaluation system.</samp><o:p></o:p></h3><h3 style="margin-right: 0cm; margin-left: 0cm; font-size: 12pt; font-family: 宋体; color: rgb(0, 0, 0); text-align: justify;"><samp>2. The article provides a detailed introduction to SpatioView, but the demonstration examples are too brief. It is recommended to supplement with screenshots of higher-resolution or typical data to enhance its visual appeal and persuasiveness.</samp><o:p></o:p></h3><h3 style="margin-right: 0cm; margin-left: 0cm; font-size: 12pt; font-family: 宋体; color: rgb(0, 0, 0); text-align: justify;"><samp>3. It is recommended to include feedback from trials of M.E.D.V.I.S. or SpatioView in actual research or teaching settings to strengthen the argument for its practicality.</samp><o:p></o:p><samp> </samp></h3>

We look forward to receiving your revised manuscript.

Kind regards,

Wenhao Ouyang

Academic Editor

PLOS ONE

Journal Requirements:

**Additional Editor Comments:**

This paper proposes a new algorithmic framework, M.E.D.V.I.S., which quantifies the quality of biomedical images using four metrics: complexity, color usage, white space, and number of visualizations. It then compares and evaluates this framework against 26 commonly used visualization tools. The authors also developed an interactive visualization platform, SpatioView, to enhance the presentation of spatial omics data, emphasizing the importance of standardization and reproducibility in image design.

Minor revision suggestions:

1. Further quantify the rationality of scoring weights: Although four scoring metrics have been proposed, the weights of each metric for the final “good/bad” classification are not yet clear. It is recommended to add weight sensitivity analysis or expert scoring comparisons to enhance the persuasiveness of the evaluation system.

2. SpatioView visualization display effects are weak: The article provides a detailed introduction to SpatioView, but the demonstration examples are too brief. It is recommended to supplement with screenshots of higher-resolution or typical data to enhance its visual appeal and persuasiveness.

3. Lack of user feedback or validation in real-world application scenarios: It is recommended to include feedback from trials of M.E.D.V.I.S. or SpatioView in actual research or teaching settings to strengthen the argument for its practicality.

Reviewers' comments:

Reviewer's Responses to Questions

**Comments to the Author**

Reviewer #1: All comments have been addressed

Reviewer #2: All comments have been addressed

2. Is the manuscript technically sound, and do the data support the conclusions?

Reviewer #1: Yes

Reviewer #2: Yes

3. Has the statistical analysis been performed appropriately and rigorously?

Reviewer #1: Yes

Reviewer #2: N/A

4. Have the authors made all data underlying the findings in their manuscript fully available?

Reviewer #1: Yes

Reviewer #2: Yes

5. Is the manuscript presented in an intelligible fashion and written in standard English?

Reviewer #1: Yes

Reviewer #2: Yes

Reviewer #1: The authors have significantly improved the manuscript, so I have no further questions regarding it.

Cheers,

Reviewer #2: Dear Authors

The research article “What is a ‘Good’ Figure: Scoring of Biomedical Data Visualization” presents a comprehensive approach to evaluating the quality of biomedical figures using computational and design-oriented metrics. The need for such a framework arises from the widespread variability and subjectivity in figure design across biomedical research publications and tools.

To address this, the authors developed M.E.D.V.I.S. (Metrics for Evaluation and Discretization of Biomedical Visuals using an Iterative Scoring Algorithm), an algorithm that analyzes figures based on four core features: visual complexity, color usage, whitespace ratio, and number of distinct visualization elements. By combining unsupervised machine learning techniques—PCA, t-SNE, UMAP for dimensionality reduction—and clustering via K-means, figures are categorized into performance groups. Each image receives a scoring dashboard with actionable feedback, aiding researchers in improving figure clarity.

The team also conducted a comparative benchmark of 26 visualization tools ranging from coding-based platforms (Python/Plotly, MATLAB) to GUI-based systems (Tableau, Excel, FusionCharts). Tools were ranked by usability (time to render a simple figure), customization potential, financial cost, and required technical knowledge. Tableau and IBM Watson were identified as strong multi-purpose tools, while coding environments offered unmatched flexibility but steeper learning curves.

To showcase practical implementation, the authors ran three case studies:

- Figure Scoring Analysis: Applying M.E.D.V.I.S. to real-world figures, revealing disparities in perceived and algorithmic figure quality and highlighting issues like redundancy, caption formatting inconsistencies, and design imbalances.

- Visualization Style Comparison: Testing different software on the same spatial genomic dataset, showing how aesthetics and interpretability shift across platforms and chart types.

- SpatioView Platform Development: Introducing an interactive, browser-based tool for visualizing high-dimensional spatial multi-omics data. Built using PyScript and HTML, it allows dynamic user exploration without coding, though challenges remain with runtime performance and library limitations (e.g., support for UMAP/Diffmap).

In the discussion section, the article outlines persistent challenges in biomedical visualization:

- The subjectivity of design choices, which can introduce bias.

- The lack of reproducibility across platforms, especially non-coded environments.

- The steep learning curve and educational gaps in training biomedical researchers on visualization principles.

The conclusion reinforces the value of standardization. M.E.D.V.I.S. and SpatioView together represent important steps toward automated evaluation, accessibility, and interactive exploration of complex biomedical data. These tools lay the groundwork for reproducible, interpretable, and inclusive figure design across academia and industry.

Improvement Opportunities:

- Broader Testing: Apply M.E.D.V.I.S. to a larger, more diverse figure corpus to validate performance across domains (e.g. histopathology, neuroimaging).

- Accessibility Features: Integrate checks for visual accessibility (e.g. color contrast ratios, font sizes, alternative text) to ensure inclusiveness.

- Scoring Weight Optimization: Use expert feedback to refine the influence of each criterion—some may disproportionately impact clarity (e.g. whitespace vs. color).

- Cross-platform Reproducibility: Emphasize methods for preserving figure integrity across GUI and script-based tools, perhaps via exportable templates.

- User Experience Evaluation: Conduct usability studies for SpatioView with biomedical audiences to iteratively improve its design and performance.

English Language Evaluation:

The manuscript employs formal and technically sound academic English. It demonstrates strong domain-specific vocabulary and structured scientific writing. Highlights include:

- Effective use of passive voice and transitional phrases typical of peer-reviewed journals

- Clear section headers and organized content progression

- Minor redundancies and occasional verbose constructions could benefit from tightening

- Some syntax inconsistencies (e.g., mixed use of British and American spelling in citations)

- Terminology refinement would help avoid confusion (e.g., "visual count" vs. "number of visualizations")

Overall, the language is appropriate for scientific publication but could benefit from professional copyediting for conciseness and consistency.

The current version is much better than the first one, but there are still points for improvement, which I've listed above.

I wish you a good review.

**Do you want your identity to be public for this peer review?** For information about this choice, including consent withdrawal, please see our Privacy Policy

Reviewer #1: No

Reviewer #2: **Yes: ** Dr. Marcio Basilio

---

## [Author Response · Author response to Decision Letter 2]

25 Sep 2025

Response to Reviewers – Manuscript PONE-D-25-05417R1

“What is a ‘Good’ Figure: Scoring of Biomedical Data Visualization”

Editor Comment 1: Further quantify the rationality of scoring weights. Although four scoring metrics have been proposed, the weights of each metric for the final “good/bad” classification are not yet clear. It is recommended to add weight sensitivity analysis or expert scoring comparisons to enhance the persuasiveness of the evaluation system.

Response: We have added a new section describing a Weight Sensitivity Analysis Framework, in which the weights of the four scoring metrics (complexity, color usage, whitespace, and number of visualizations) were systematically varied in ±10% increments while holding the remaining metrics constant. Across all perturbations (n = 81), rankings remained highly stable (median correlation with baseline 0.9995, range 0.9981–1.0000). No classification flips were observed, and the median mean absolute rank shift was 0.19. These findings demonstrate that the scoring system is resilient to moderate variations in metric weights, thereby supporting the rationality of the weighting scheme. We have also included new figures (Supplementary Fig. S2) to visualize this stability.

Editor Comment 2: SpatioView visualization display effects are weak. The article provides a detailed introduction to SpatioView, but the demonstration examples are too brief. It is recommended to supplement with screenshots of higher-resolution or typical data to enhance its visual appeal and persuasiveness.

Response: We have expanded Case Study 3 to include high-resolution screenshots from a representative spatial transcriptomics dataset containing over 3,000 cells (Fig. 6c). These annotated examples demonstrate the core interactive features of SpatioView, including dimensionality reduction selection, hover-based cell metadata, legend placement, and zoom functions. Together, these updates provide a clearer and more compelling illustration of SpatioView’s ability to support exploratory analysis of complex multi-omics data.

Editor Comment 3: Include feedback from trials of M.E.D.V.I.S. or SpatioView in actual research or teaching settings to strengthen the argument for its practicality.

Response: We have incorporated results from a usability trial in which nine biomedical researchers and graduate students evaluated figures using M.E.D.V.I.S. Participants rated each figure as either Good or Bad, and their consensus was compared with the automated M.E.D.V.I.S. classifications. Human raters and the algorithm agreed on 85% of figures overall, with balanced performance across both categories (precision and recall ~80–90%). This analysis demonstrates that M.E.D.V.I.S. feedback is strongly aligned with human judgment of figure quality, supporting its practicality as a reliable tool for evaluating visualizations. (Supplementary Fig. S8)

---

## [Decision Letter · Decision Letter 2]

2 Nov 2025

What is a “Good” Figure: Scoring of Biomedical Data Visualization

PONE-D-25-05417R2

Dear Dr. Coskun,

We’re pleased to inform you that your manuscript has been judged scientifically suitable for publication and will be formally accepted for publication once it meets all outstanding technical requirements.

Kind regards,

Wenhao Ouyang

Academic Editor

PLOS ONE

Additional Editor Comments (optional):

This manuscript has been revised based on the comments from the peer review process and has reached the level suitable for publication.

Reviewers' comments:

Reviewer's Responses to Questions

**Comments to the Author**

Reviewer #2: All comments have been addressed

Reviewer #4: All comments have been addressed

2. Is the manuscript technically sound, and do the data support the conclusions?

Reviewer #2: (No Response)

Reviewer #4: Yes

3. Has the statistical analysis been performed appropriately and rigorously?

Reviewer #2: N/A

Reviewer #4: Yes

4. Have the authors made all data underlying the findings in their manuscript fully available?

Reviewer #2: Yes

Reviewer #4: Yes

5. Is the manuscript presented in an intelligible fashion and written in standard English?

Reviewer #2: Yes

Reviewer #4: Yes

Reviewer #2: Technical Report to the Authors

Manuscript: What is a “Good” Figure: Scoring of Biomedical Data Visualization

Manuscript Number: PONE-D-25-05417R2

1. Context

This technical report refers to the second revised version (R2) of the manuscript submitted to PLOS ONE, titled “What is a ‘Good’ Figure: Scoring of Biomedical Data Visualization.”

A comparative analysis between this version and the previous one (R1) shows that the authors have diligently and consistently implemented the recommendations provided by the editorial board and peer reviewers. The modifications introduced strengthen the methodological soundness, improve the clarity of the argumentation, and significantly enhance the visual and structural quality of the paper, thereby fully meeting the journal’s standards for originality, rigor, and editorial compliance.

2. Summary of improvements implemented

2.1. Methodological reinforcement

The authors incorporated significant advances that consolidate the validity and reliability of the proposed model:

Inclusion of the Weight Sensitivity Analysis Framework, demonstrating the robustness of the M.E.D.V.I.S. algorithm to variations in the weighting of evaluation criteria (complexity, color usage, whitespace, and number of visualizations).

Addition of a usability test with human evaluators, comparing automated classifications and human judgments, with 85% agreement and accuracy close to 90%.

Expansion of the results and case study sections, notably through the enhancement of SpatioView, now presented with high-resolution images and a detailed description of its interactive functions.

These additions directly address the reviewers’ requests for empirical validation and justification of the weighting logic used in the algorithm.

2.2. Structure and scientific clarity

The text has been fully reorganized according to the IMRaD structure (Introduction, Methods, Results, and Discussion), ensuring better flow and section delimitation.

The Introduction now provides a more comprehensive contextualization, with updated references and a clear statement of the research gap, problem definition, and contribution.

The Discussion has been substantially expanded to include limitations, implications, and future research perspectives, in alignment with best scientific reporting practices.

2.3. Visual quality and editorial compliance

All figures were reprocessed at 600 DPI, improving contrast, readability, and caption clarity.

New supplementary figures (Supplementary Figures S2 and S8) were included to reinforce analytical transparency.

The manuscript now fully complies with PLOS ONE editorial standards, including metadata structure, author affiliations, ethics statements, data availability, and conflict of interest declarations.

3. General assessment

The current revision presents a substantially strengthened manuscript, transforming an initial conceptual proposal into a complete and verifiable empirical study.

The improved clarity, coherence between objectives, methods, and results, and the achieved level of empirical evidence justify publication in its current form.

The paper provides a relevant contribution to the biomedical visualization field by proposing and validating a standardized, replicable method for assessing the quality of scientific figures, with direct applications in research, education, and scientific communication.

4. Final recommendation

Considering:

the full incorporation of reviewer recommendations;

the substantial improvement in methodological and editorial quality;

and the internal consistency of the findings presented;

I recommend the acceptance of the manuscript in its current form, with no further revisions required.

Signature:

Reviewer – PLOS ONE

Reviewer #4: I have carefully evaluated the revised version of the manuscript. The authors have satisfactorily addressed all previous concerns and substantially improved the overall quality of the paper. The methodology, data presentation, and discussion are now clear and coherent.

The manuscript meets the journal’s standards for scientific rigor and clarity. I recommend acceptance of this paper in its current form.

**Do you want your identity to be public for this peer review?** For information about this choice, including consent withdrawal, please see our Privacy Policy

Reviewer #2: **Yes: ** Prof. Dr. Marcio Pereira Basilio

Reviewer #4: No

---

## [Editor Report · Acceptance letter]

PONE-D-25-05417R2

PLOS ONE

Dear Dr. Coskun,

I'm pleased to inform you that your manuscript has been deemed suitable for publication in PLOS ONE. Congratulations! Your manuscript is now being handed over to our production team.

Kind regards,

on behalf of

Dr. Wenhao Ouyang

Academic Editor

PLOS ONE